# VISUALOVERLOAD: PROBING VISUAL UNDERSTANDING OF VLMS IN *Really* DENSE SCENES

## ABSTRACT

Is basic visual understanding really solved in state-of-the-art VLMs? We present VisualOverload, a slightly different visual question answering (VQA) benchmark comprising 2,720 question–answer pairs, with privately held ground-truth responses. Unlike prior VQA datasets that typically focus on near global image understanding, VisualOverload challenges models to perform simple, knowledge-free vision tasks in densely populated (or, *overloaded*) scenes. Our dataset consists of high-resolution scans of public-domain paintings that are populated with multiple figures, actions, and unfolding subplots set against elaborately detailed backdrops. We manually annotated these images with questions across six task categories to probe for a thorough understanding of the scene. We hypothesize that current benchmarks overestimate the performance of VLMs, and encoding and reasoning over details is still a challenging task for them, especially if they are confronted with densely populated scenes. Indeed, we observe that even the best model (*o3*) out of 37 tested models only achieves $19.8\%$ accuracy on our hardest test split and overall $69.5\%$ accuracy on all questions. Beyond a thorough evaluation, we complement our benchmark with an error analysis that reveals multiple failure modes, including a lack of counting skills, failure in OCR, and striking logical inconsistencies under complex tasks. Altogether, VisualOverload exposes a critical gap in current vision models and offers a crucial resource for the community to develop better models.

**Dataset and Leaderboard:** *(hidden during the review)*[1]

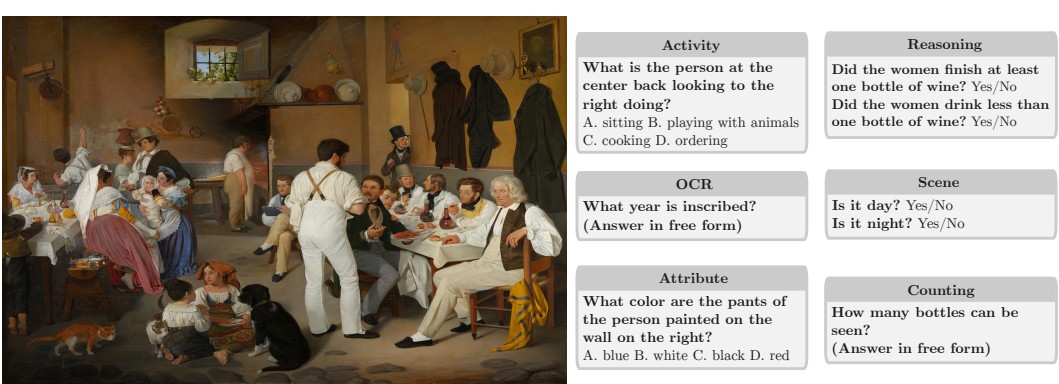

Figure 1: **Example questions from VisualOverload.** Our benchmark consists of images displaying densely populated scenes paired with handcrafted questions (multiple-choice and free-form) covering six core vision tasks. All yes/no questions are paired with questions asking for a logical opposite question to decrease the random chance and to provide an additional signal for measuring *logical consistency*.

---

[1]The dataset, a leaderboard, and evaluation server will be hosted on HuggingFace. The dataset is temporarily hosted at https://anonymous.4open.science/r/iclr26-visualoverload-6442.

# 1 INTRODUCTION

Visual question answering (VQA) (Antol et al., 2015; Goyal et al., 2017; Agrawal et al., 2018) has emerged as a common benchmark for image understanding in VLMs. Recent state-of-the-art models achieve surprisingly strong results on established VQA datasets (Li et al., 2023b; Yu et al., 2024), suggesting that basic forms of visual understanding might already be "solved". In turn, several benchmarks have shifted from generic image understanding towards the probing of domain-specific knowledge (Yue et al., 2024; Phan et al., 2025).

However, *have today's VLMs really solved core vision tasks?* We argue that current benchmarks are poor indicators of this, as most of them fail to capture the complexity of real-world applications, where safety and reliability depend on fine-grained perception in dense and high-resolution scenes. Current benchmarks instead emphasize simple foreground reasoning (Li et al., 2023b; 2024b; Yu et al., 2024) or needle-in-a-haystack-like retrieval tasks (Wu & Xie, 2024; Li et al., 2023a; Shi et al., 2025), falling short of testing such capabilities, and potentially overestimating performance.

Instead, we expect that model performance will severely drop "under pressure", and modulate this through the angle of visual complexity and dense–*visually overloaded*–scenes. We motivate our analysis by suggesting that the vision encoder is a bottleneck in modern VLMs. Encoders are designed to compress visual input into a fixed number of tokens, retaining only the most salient features. This design imposes an inherent upper bound on fine-grained perception: for instance, a ViT-L/14@336px encoder maps $336^2 \times 3$ pixels into just 1024 tokens, inevitably discarding information. While random noise illustrates an extreme case of this, we expect sufficiently densely populated scenes to already trigger these limits.

To verify our expectations, we introduce a new dataset explicitly designed to probe image understanding in dense and high-resolution scenes. Our dataset comprises 150 high-resolution scans of artworks featuring highly dense scenes, along with 2,720 manually curated question–answer pairs spanning six fundamental tasks of visual comprehension: activity recognition, attribute recognition, counting, optical character recognition (OCR), visual reasoning, and global scene classification (see Fig. 1 for an example). Unlike prior benchmarks that recycle existing image datasets, all of our images are newly sourced from public domain artworks, resulting in a fresh source of data free of copyright concerns.

Our empirical study of 37 VLMs reveals that state-of-the-art models, while often competent at global scene classification, consistently struggle in fine-grained recognition in dense scenes. To better characterize these challenges, we split our benchmark into three difficulty levels (easy, medium, hard), calibrated by average model performance. Even the strongest model we tested (o3) achieves only 19.8% accuracy on the hardest split and 69.5% overall, underscoring the difficulty of the benchmark and the underlying challenge.

Finally, we conduct a detailed error analysis and uncover striking failures: for instance, we observe strong failures in counting tasks for high ground-truth values and in OCR tasks requiring precise textual recognition, such as the recognition of typos. Furthermore, we observe that models frequently provide logically inconsistent answers to logically opposite paired questions, with this instability intensifying as the complexity of such queries increases. Such inconsistencies sometimes even degrade performance to random or even sub-random baselines, suggesting that these models rely heavily on shortcuts rather than robust reasoning. Taken together, these findings highlight the urgent need for benchmarks like ours that reflect the realities of dense, high-resolution perception and reveal fundamental limitations of current VLMs.

We summarize our contributions as follows:

- We introduce a new benchmark for VQA in dense, high-resolution (*visually overloaded*) scenes. Our benchmark contains 2,720 manually curated question–answer pairs across six fundamental categories (activity recognition, attribute recognition, counting, OCR, visual reasoning, and global scene classification) as described in Sec. 2. Ground truths are held private to avoid target leakage. All images are sourced entirely from public domain artwork collections to provide a fresh image dataset free of copyright issues.

- We evaluate a range of state-of-the-art models in Sec. 3 and show that, while they perform well on global scene classification, they struggle significantly in fine-grained understanding

in dense settings, particularly for counting and OCR. We provide a three-level difficulty split, calibrated by average model performance, showing that even the strongest tested model (o3) reaches only $19.8\%$ accuracy on the hardest split.

- We perform a detailed error analysis in Sec. 4, uncovering systematic inconsistencies and shortcut biases that further hinder robust performance in visually overloaded settings.

## 2 THE VISUALOVERLOAD BENCHMARK

Our goal is to create a benchmark that tests basic image recognition skills that we expect to be present in any frontier models. However, unlike many previous benchmarks, we design our benchmark around fine-grained recognition in dense scenes to stress test the vision encoders' representation. In the following subsections, we discuss the dataset curation (Sec. 2.1), evaluation process (Sec. 2.2), and discuss differences to other benchmarks in detail (Sec. 2.3).

### 2.1 DATASET CURATION

**Image collection.**   We collected 150 high-resolution digitizations of paintings, curated from collections held by museums around the world and made available through Google Arts & Culture. We specifically selected paintings that depict visually complex scenes — densely composed narratives filled with numerous figures, actions, and subplots, often unfolding simultaneously within richly detailed environments. While complexity is hard to quantify, we picked artworks that tend to overwhelm the eye and demand significant time and attention to fully absorb their intricate details, as a rule of thumb. We only selected paintings in the public domain, *i.e.*, artworks where the original creators passed away more than 100 years ago.

Due to the inherent complexity of the scenes, the images in the dataset are typically of extreme resolution and exceed 4K resolution ($3840 \times 2160$ pixels). To standardize the dataset, we downsampled all images to match the nearest total pixel count of 4K while preserving their original aspect ratios. 28 images were originally below 4K resolution and were therefore not downsampled; however, all remain above Full HD resolution ($1920 \times 1080$ pixels).

**Question annotation.**   Six human annotators manually annotated the resized images with questions and answer options. The annotators were instructed to generate questions that are clearly formulated and specific, leaving no ambiguity about the information being requested. To avoid language priors, the questions are also explicitly mandated to be grounded in the content of the accompanying image and should not be answered from text alone (Zhang et al., 2016; Goyal et al., 2017; Agrawal et al., 2016; 2018; Cadene et al., 2019). In addition, we restricted questions to probe for details that can be directly observed or reasonably inferred from the image, excluding any question–answer pairs based on beliefs or subjective interpretations. Finally, we requested questions to be solvable without external or expert knowledge beyond a basic level of everyday "world" knowledge, as we are only concerned with image understanding in this work.

We employ two answer formats: multiple-choice and freeform. Multiple-choice questions either offer four options, where only one correct answer or are binary yes/no questions. We pair each of the latter kind of questions with a logical opposite (*e.g.*, *"Is it day?"* and *"Is it night?"*) (Zhang et al., 2016). This not only helps calibrate against random guessing but also provides an additional signal for identifying logical inconsistencies in generated responses (see Sec. 4). For selected tasks, we use freeform answers to raise the level of difficulty (see below).

Our annotated questions each fall into one of the following six categories, resulting in approximately 18 questions per image:

- **Activity recognition** ($N = 150$): multiple-choice questions about actions or activities occurring in the scene. These questions will refer to a single or a group of subjects, typically paired with a constraint. For instance, *"What is the person dressed in brown at the front of the table in the leftmost house doing?"*.
- **Attribute recognition** ($N = 149$): multiple-choice queries about the color attribute of objects are typically paired with a constraint probing for spatial, attribute, or activity recognition. For instance, *"What is the color of the left-most ship flag?"*.

- **Counting** ($N = 559$)**:** freeform inquiries about details that involve determining the number of objects present. The questions may be related to the entire scene or spatially constrained, requiring mild visual reasoning to provide a correct answer. For instance, *"How many roses are lying on the floor?"*.

- **OCR** ($N = 118$)**:** freeform queries about written text in the image. Languages include English, Latin, Chinese, Dutch, and Greek. Some questions are specified to probe for parts of the text, which can be seen as a mild form of text reasoning, *e.g.*, *"What is the last name of the signature?"*, or require some minimal visual reasoning efforts, *e.g.*, *"What does the word below the main character read?"*.

- **Reasoning** ($N = 356$)**:** multiple-choice queries that require a medium to high load of visual reasoning to be answerable. In principle, we expect that a "chain of thought" is necessary to provide a correct answer. For instance, these questions may require functional or intent understanding, distance or path estimation, light- or wind-source estimation, occupancy detection, and numerical comparisons based on the image's content. Some example questions are: *"Do you have to cross the water to reach the two windmills on the right?"*, *"I am allergic to seafood, is all of the food on the table safe for me?"*, or *"Does capital punishment appear to be legal in this scene?"*.

- **Scene classification** ($N = 1388$)**:** multiple-choice queries about the overall scene or setting of the image. These questions typically do not require a fine-grained understanding or complex visual reasoning of the scene, and we expect all models to perform well on them. Yet, we still observe that some models struggle with them. For instance, *"Are there animals in the scene?"*.

**Quality control.** After annotation, we evaluated 37 VLMs on our dataset and manually verified the correctness of ground truths if the question was only solved by a small number of models. Furthermore, we evaluated the performance of 3 of the strongest models from our leaderboard (InternVL3-38B, Qwen2.5-VL 32B, LLaVA-OV 72B) on our dataset while ablating the image to probe for hidden biases due to linguistic cues in the question or answer options of multiple-choice questions. We detected a number of questions where all 3 models were able to answer the question without seeing the image. We then prompted Gemini 2.5 Pro to detect language biases in each question (see appendix A.3 for the prompt) and removed instances with severe biases, such as cases where the correct answer was an oddity or was implied by the context of the question. Please note that this is not necessary for freeform answers (counting, OCR) or binary questions, which are self-balanced by their logical opposites. The final "blind" performance on the remaining questions is shown in Tab. 1. Overall, our quality control resulted in a reduction of blind performance to near chance baselines for most tasks. However, we still observe elevated performance for the attribute recognition and counting tasks. These gains stem primarily from statistical irregularities in the distribution of ground-truth answers (*e.g.*, small object counts being more frequent). Such distributional priors are unavoidable in real-world datasets and do not confer a generalizable shortcut that undermines evaluation. In practice, models must still extract and process visual content to achieve strong performance on all of our tasks.

**Difficulty splits.** We divide our questions into three difficulty levels—easy, medium, and hard—based on model performance in Sec. 3. The thresholds are defined by the percentage of correct responses: $[0, 20]$ for hard, $(20, 90)$ for medium, and $[90, 100]$ for easy.

### 2.2 Evaluation Process

**Metrics.** We rely on the average accuracy as the principal metric for our benchmark, scored over all questions, each difficulty split, as well as each task category. We define an answer as accurate if it precisely matches the ground truth label. For binary questions, we measure pair-wise accuracy, and score a pair as correct if both questions are correct, and false otherwise.

**Answer extraction.** Although our prompts aim to constrain output format, VLMs do not always follow these instructions. To address this, we apply simple heuristic-based preprocessing to extract and normalize responses across tasks.

For multiple-choice questions, we detect the option letter and map it to the corresponding label, or directly match the label when possible. For counting questions, we extract either numeral or

Table 1: **Blind benchmark results.** We benchmark three models on VisualOverload without the images to measure a potential language bias.

| Model | Params [B] | Activity (150) | Attributes (149) | Counting (559) | OCR (118) | Reasoning (356) | Scene (1388) | Easy (986) | Medium (1304) | Hard (430) | Total (2720) |
|---|---|---|---|---|---|---|---|---|---|---|---|
| *Random Chance* | - | 25.0 | 25.0 | 0.0 | 0.0 | 25.0 | 25.0 | 24.5 | 16.7 | 3.7 | 16.0 |
| *Consistent Chance* | - | 25.0 | 25.0 | 0.0 | 0.0 | 42.5 | 50.0 | 47.2 | 26.2 | 4.7 | 27.2 |
| InternVL3 38B | 38 | 30.0 | 34.9 | 15.6 | 0.8 | 36.6 | 24.2 | 57.3 | 40.8 | 9.5 | 22.8 |
| Qwen2.5-VL 32B | 32 | 32.0 | 26.2 | 8.8 | 0.0 | 29.3 | 38.0 | 47.6 | 37.8 | 7.7 | 24.5 |
| LLaVA-OV 72B | 72 | 29.3 | 40.3 | 18.1 | 0.8 | 36.1 | 38.6 | 38.2 | 51.1 | 7.9 | 29.2 |

lexical integer forms, defaulting to the last-mentioned integer if multiple candidates appear. For OCR tasks, we extract the relevant text, then normalize it by removing diacritics, punctuation, and spacing, converting to lowercase, and replacing 'V' with 'U' and 'J' with 'I' to reduce ambiguity in Latin texts.

**Evaluation server.**   To prevent test leakage into future VLMs, we hold out the ground truth and only release the image samples and questions. We do not provide a development split, as our tasks do not require any specialized knowledge or skills, and we expect decent foundational vision models to solve these tasks without finetuning. Instead, we provide an evaluation server that scores generated answers and maintain an opt-in leaderboard of those. Evaluations are made by submitting a JSON file with model predictions to our public evaluation server. The server applies our extraction heuristics as outlined above, but users are free to apply their own preprocessing of any kind before submitting their predictions. We rate-limit the server per user and day to prevent ground-truth extraction attacks.

## 2.3 COMPARISON WITH EXISTING BENCHMARKS

Existing VQA benchmarks underestimate the true difficulty of visual reasoning. They rely on low-resolution images, recycled content, and automatically generated questions that encourage shallow pattern matching rather than genuine scene understanding. Our benchmark is intentionally designed to correct these shortcomings and to set a higher standard for evaluation. Its distinguishing features are:

- **High-resolution, dense images.** We collect detailed images of complex scenes, enabling questions that demand fine-grained perception and long-range reasoning. Unlike prior benchmarks, which often reduce vision to global features, our dataset forces models to engage with the full richness of the scene.

- **Manual annotation.** All questions are crafted by human annotators. Automated pipelines used in other datasets may scale cheaply, but they also introduce biases, trivial patterns, and low-quality queries. Our human-centered approach ensures natural, challenging, and unbiased evaluation.

- **Fresh image data.** Rather than recycling existing dataset sources, we provide entirely new images. This prevents leakage from pretraining corpora and eliminates the domain biases that plague benchmarks built from reused datasets.

- **Public domain licensing.** Every image is sourced from the public domain, removing legal barriers that limit distribution or usage. Unlike benchmarks with restrictive or unclear licensing due to web crawling, ours is openly and universally accessible.

In sum, where existing benchmarks compromise on difficulty, reliability, or ethics, our dataset sets a new bar: more challenging, more trustworthy, and more responsible. It is not simply another addition to the landscape, but a necessary corrective to the limitations of current VQA evaluation.

## 3 EXPERIMENTS

In the following subsections, we evaluate the performance of different VLMs on VisualOverload. In Sec. 3.1 we introduce the models, and assess their performance in Sec. 3.2.

## 3.1 BASELINES

We evaluate 37 recent VLMs, including variously sized open-weight models ranging from 450M to 109B parameters, designed for low- and high-resolution image understanding, that we separate into three parameter bands, specialized high-resolution understanding models, and 4 proprietary frontier models. To simplify the answer extraction, we add small postfixes to the benchmark questions outlined in appendix A.1. We generate answers using greedy decoding for all models, except for proprietary models and models where greedy decoding failed to generate useful outputs (*e.g.*, Llama 4), as highlighted in the result tables.

Additionally, we compare the results to *random chance* (we assume no priors for counting and OCR), as well as *consistent chance*, where we assume that a model is guessing, but gives consistent guesses for logically opposite questions.

## 3.2 MAIN RESULTS

The results in Tab. 2 show vast differences between models and some of the tasks in VisualOverload. First off, we notice that all models struggle with our freeform counting and OCR tasks. The best accuracy in counting is achieved by Gemini 2.0 Flash, but is only at $41.7\%$. OCR performance is overall better, but even the best model, o4-mini only achieves $62.7\%$. This is also the task with the highest discrepancy between proprietary commercial models and open-weight ones.

For activity and attribute recognition, we see an improved accuracy (yet, also a higher random chance), but still far from satisfactory performance even with the strongest models. For reasoning tasks, we find that almost all models struggle and make rather small improvements compared to the consistent random chance, while some of the smaller models even underperform it. The only positive outlier here is o3, which achieves a significant advantage compared to other models, presumably due to its reasoning mode. Unsurprisingly, we find that frontier models achieve a high accuracy on scene understanding, as it primarily relies on a superficial understanding of the scenes, as is common in many of the existing VQA datasets. However, rather surprisingly, the task can still be challenging for many other models, even for large models. Yet, 8B parameters seem already to be sufficient to achieve $93.4\%$. In a few cases, the accuracy even fell below a consistent chance, suggesting a fallback to shortcut features (see also Sec. 4).

Averaged over all tasks, the best model (o3) achieves only $19.8\%$ on the hardest test split, and $69.5\%$ overall. The strongest open-weight model is InternVL3 38B with $7.2\%$ and $67.6\%$, respectively. Interestingly, we found that specialized HD models perform significantly worse than equally sized regular models. We attribute this primarily to the fact that most VLMs apply methodologies such as AnyRes (Liu et al., 2024a) to support high-resolution images and, thus, performance is rather dependent on the backbones and training, therefore showing that modern VLMs outperform specialized VLMs built on older backbones[2]. Finally, we also find some counter-intuitive scaling trends, where performance decreases with parameter size (often for the largest model of the family, *i.e.*, in InternVL3 and PaliGemma 2).

We encourage the community to explore advanced prompting techniques and invite them to submit these to our leaderboard.

## 4 ERROR ANALYSIS

In this section, we aim to better outline the errors that models make. With the protection of our private ground-truth in mind, we will rely on average statistics over all models described in Sec. 3.1.

**Counting.** To analyze errors in counting tasks, we plot the distribution of predictions versus ground truths in Fig. 2a. Models are generally accurate when the ground truth is low, but errors increase substantially as the ground truth rises. Although some errors stem from incorrect predictions, many are also due to refusals (which we treat as 0) or blank responses (*e.g.*, "too many objects to count"). In all cases, models tend to err on the low side and underestimate the ground truth. Yet, our analysis also contained outliers showing severe overestimation.

---

[2]Please find an ablation of performance under different resolutions of VisualOverload in appendix A.2.

Table 2: **Benchmark results.** We report the accuracy as the fraction of correct responses after processing, including the accuracy normalization for binary questions for each of the categories in our benchmark, as well as the average. **Legend:** [S] Completions were generated using stochastic sampling at default parameters.

| Model | Params [B] | Activity (150) | Attributes (149) | Counting (559) | OCR (118) | Reasoning (356) | Scene (1388) | Easy (986) | Medium (1304) | Hard (430) | Total (2720) |
|---|---|---|---|---|---|---|---|---|---|---|---|
| *Random Chance* | - | 25.0 | 25.0 | 0.0 | 0.0 | 25.0 | 25.0 | 24.5 | 16.7 | 3.7 | 16.0 |
| *Consistent Chance* | - | 25.0 | 25.0 | 0.0 | 0.0 | 42.5 | 50.0 | 47.2 | 26.2 | 4.7 | 27.2 |
| *Small Open-Weight Models (< 7B)* | | | | | | | | | | | |
| PaliGemma 2 3B (Steiner et al., 2024) | 3.0 | 42.0 | 53.0 | 20.4 | 8.5 | 24.9 | 32.7 | 78.7 | 42.6 | 6.3 | 29.0 |
| LLaVA 1.5 7B (Liu et al., 2023a) | 7.0 | 35.3 | 43.6 | 13.2 | 3.4 | 39.5 | 43.2 | 87.3 | 35.1 | 2.3 | 30.8 |
| Gemma 3n E2B (Gonzalez & Shivanna, 2025) | 5.0 | 32.0 | 26.2 | 15.0 | 19.5 | 35.6 | 53.2 | 83.2 | 39.9 | 8.6 | 33.9 |
| LLaVA-NeXT 7B (Liu et al., 2024a) | 7.0 | 44.7 | 41.6 | 19.1 | 8.5 | 40.5 | 54.0 | 93.2 | 39.5 | 2.6 | 37.5 |
| LFM2 VL 450M (Liquid AI, 2025) | 0.4 | 35.3 | 47.0 | 22.9 | 20.3 | 27.8 | 59.5 | 94.5 | 43.9 | 8.6 | 39.7 |
| DeepSeek VL2 Tiny (Wu et al., 2024) | 1.0 | 54.7 | 47.7 | 22.5 | 35.6 | 37.1 | 54.2 | 88.8 | 45.8 | 4.4 | 41.2 |
| SmolVLM (Marafioti et al., 2025) | 2.0 | 42.7 | 41.6 | 17.2 | 28.0 | 32.2 | 67.3 | 96.3 | 47.5 | 4.2 | 42.0 |
| Gemma 3n E4B (Gonzalez & Shivanna, 2025) | 5.0 | 40.0 | 23.5 | 19.3 | 23.7 | 41.0 | 73.9 | 92.7 | 50.5 | 9.1 | 44.2 |
| InternVL3 1B (Zhu et al., 2025) | 1.0 | 48.0 | 57.0 | 27.2 | 25.4 | 35.1 | 77.5 | 98.7 | 57.9 | 5.6 | 50.6 |
| LFM2 VL 1.6B (Liquid AI, 2025) | 1.6 | 49.3 | 55.7 | 25.2 | 28.0 | 44.4 | 79.5 | 99.3 | 60.1 | 5.1 | 51.9 |
| Qwen2.5-VL 3B (Bai et al., 2025) | 3.0 | 60.7 | 61.7 | 25.9 | 49.2 | 43.9 | 77.5 | 98.9 | 61.9 | 4.9 | 54.1 |
| InternVL3 2B (Zhu et al., 2025) | 2.0 | 50.0 | 58.4 | 30.4 | 39.0 | 49.8 | 80.3 | 99.9 | 62.6 | 6.3 | 55.3 |
| DeepSeek VL2 (Wu et al., 2024) | 4.5 | 65.3 | 63.8 | 25.9 | 46.6 | 58.5 | 81.8 | 99.9 | 66.7 | 4.0 | 57.7 |
| *Medium Open-Weight Models (7—13B)* | | | | | | | | | | | |
| LLaVA-OV 7B (Li et al., 2024a) | 7.0 | 60.7 | 57.7 | 28.4 | 29.7 | 54.1 | 88.2 | 99.6 | 68.0 | 4.7 | 58.3 |
| Qwen2.5-VL 7B (Bai et al., 2025) | 7.0 | 63.3 | 69.1 | 34.9 | 55.9 | 49.8 | 85.3 | 99.8 | 71.2 | 9.3 | 61.5 |
| LLaVA 1.5 13B (Liu et al., 2023a) | 13.0 | 41.3 | 39.6 | 13.8 | 3.4 | 42.9 | 71.6 | 95.6 | 43.5 | 2.8 | 42.0 |
| LLaVA-NeXT 13B (Liu et al., 2024a) | 13.0 | 44.0 | 43.6 | 17.0 | 6.8 | 41.5 | 75.8 | 99.0 | 46.9 | 3.7 | 45.1 |
| Gemma 3 12B (Gemma Team, 2025) | 12.0 | 48.7 | 42.3 | 16.5 | 31.4 | 47.8 | 82.7 | 99.5 | 54.5 | 6.5 | 50.0 |
| PaliGemma 2 10B (Steiner et al., 2024) | 10.0 | 48.7 | 52.3 | 23.6 | 5.1 | 42.4 | 81.8 | 98.3 | 56.3 | 6.3 | 50.3 |
| InternVL3 8B (Zhu et al., 2025) | 8.0 | 66.0 | 67.8 | 32.2 | 42.4 | 59.0 | 93.4 | 100.0 | 75.1 | 8.4 | 63.9 |
| *Large Open-Weight Models (> 13B)* | | | | | | | | | | | |
| PaliGemma 2 28B (Steiner et al., 2024) | 28.0 | 40.0 | 49.0 | 17.4 | | 40.0 | 66.1 | 92.0 | 47.9 | 8.1 | 41.5 |
| Gemma 3 27B (Gemma Team, 2025) | 27.0 | 51.3 | 46.3 | 18.1 | 40.7 | 50.7 | 86.3 | 99.6 | 57.8 | 8.8 | 53.2 |
| Llama 4 Scout (Meta AI, 2025) | 109.0 | 58.7 | 65.8 | 31.1 | 37.3 | 62.0 | 78.8 | 99.4 | 64.0 | 14.0 | 57.5 |
| InternVL3 14B (Zhu et al., 2025) | 14.0 | 66.7 | 69.1 | 30.6 | 41.5 | 57.1 | 91.1 | 99.8 | 73.5 | 5.1 | 62.5 |
| LLaVA-OV 72B (Li et al., 2024a) | 72.0 | 66.0 | 69.8 | 30.9 | 39.0 | 57.1 | 91.8 | 99.8 | 73.8 | 4.4 | 62.7 |
| Qwen2.5-VL 32B (Bai et al., 2025) | 32.0 | 60.0 | 70.5 | 30.8 | 61.0 | 61.5 | 90.3 | 99.9 | 72.6 | 13.6 | 63.6 |
| Qwen2.5-VL 72B (Bai et al., 2025) | 72.0 | 67.3 | 74.5 | 35.1 | 72.9 | 53.2 | 90.5 | 99.8 | 76.2 | 13.0 | 65.7 |
| InternVL3 78B (Zhu et al., 2025) | 78.0 | 78.0 | 80.5 | 34.7 | 31.4 | 65.4 | 93.7 | 99.7 | 80.6 | 8.1 | 66.8 |
| InternVL3 38B (Zhu et al., 2025) | 38.0 | 76.7 | 78.5 | 35.4 | 45.8 | 69.8 | 92.2 | 99.7 | 81.8 | 7.2 | 67.6 |
| *Specialized High-Resolution Models* | | | | | | | | | | | |
| VILA HD 4K[S] (Shi et al., 2025) | 8.0 | 54.0 | 48.3 | 22.5 | 11.0 | 49.3 | 74.5 | 99.0 | 55.8 | 4.0 | 48.5 |
| VILA HD 1.5K[S] (Shi et al., 2025) | 8.0 | 54.0 | 57.7 | 25.9 | 21.2 | 52.2 | 79.4 | 99.4 | 61.0 | 4.0 | 53.1 |
| ILM-XC2-4KHD (Dong et al., 2024) | 7.0 | 50.7 | 53.7 | 25.4 | 31.4 | 42.4 | 83.6 | 99.2 | 61.1 | 6.7 | 53.4 |
| ILM-XC2.5 (Zhang et al., 2024a) | 7.0 | 48.0 | 51.7 | 22.7 | 35.6 | 45.9 | 87.3 | 99.5 | 61.8 | 8.8 | 54.3 |
| *Proprietary Models* | | | | | | | | | | | |
| Horizon Alpha[S] (Horizon Alpha Team, 2025) | — | 57.3 | 74.5 | 35.6 | 48.3 | 63.9 | 93.2 | 99.7 | 76.8 | 10.7 | 65.7 |
| Gemini 2.0 Flash[S] (Gemini Team, 2025) | — | 76.0 | 71.1 | 41.7 | 57.6 | 56.6 | 92.1 | 99.5 | 77.3 | 19.5 | 68.1 |
| o4-mini[S] (OpenAI, 2025) | — | 70.0 | 76.5 | 38.3 | 62.7 | 67.8 | 93.7 | 100.0 | 80.6 | 17.2 | 69.1 |
| o3[S] (OpenAI, 2025) | — | 74.0 | 69.8 | 36.7 | 61.0 | 75.1 | 94.7 | 99.9 | 80.2 | 19.8 | 69.5 |

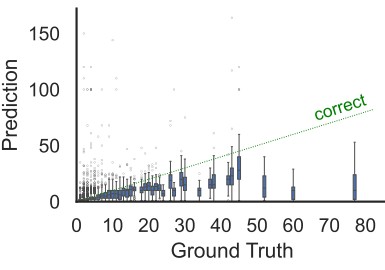

(a) **Prediction vs. ground truths.**

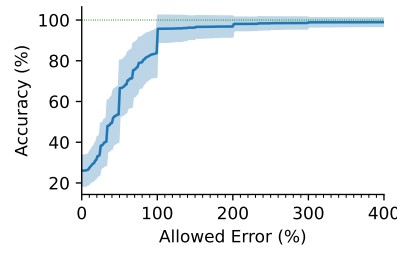

(b) **Accuracy under tolerance** (mean±std).

Figure 2: **Insights into counting errors.** All analyses display distributions over all model predictions exclusively for the counting task.

To quantify the magnitude of these errors, we measured accuracy under varying tolerance levels, shown in Fig. 2b. Prediction errors are typically severe: even with a $10\%$ tolerance, average accuracy improves by only $1.6\%$. Larger tolerances, such as $50\%$ or $100\%$, yield more substantial improvements, but such levels are impractical for real-world applications.

**OCR.** Similar to counting, we aim to quantify the magnitude of errors in OCR predictions. To do this, we measure the Levenshtein edit distance (Levenshtein, 1965) between preprocessed predictions (as described in Sec. 2.2) and ground truths for incorrect answers. We normalize the distance by the maximum sequence length and visualize the distribution in Fig. 3. The distribution's center of mass is around $0.7$, indicating that sequences require substantial edits to be correct, highlighting severe errors.

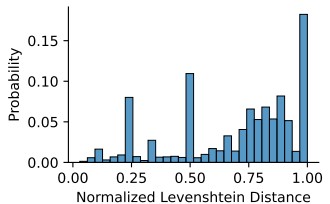

Figure 3: **OCR prediction error distance.**

Manual inspection of a subset of errors reveals three main causes: hallucinations, extraction of irrelevant text, and, in a few severe cases, failure to follow the instruction to respond only with the text. Errors of the second type often arise from misinterpretation of text flow, such as side-by-side multi-line paragraphs or non-standard layouts like banners. For errors with low edit distance, we frequently observe that models' auto-correct spelling or generally fall back to more probable token sequences rather than reproducing the actual text (*e.g.*, "accidunt" becomes "accident"), particularly in non-English or non-Latin scripts.

**Logical Consistency.** As described in Sec. 2.1, our dataset contains binary questions, where each such question is paired with a logically opposite. A strong model should argue logically consistent, even if the answer is wrong. For instance, if a model answers "yes" to "Is it day?" it should answer "no" to "Is it night?". We measure the ratio of logically consistent answer pairs per model and task (reasoning and global scene understanding) and visualize the results in Fig. 4.

We observe that frontier models answer fairly logically consistent for the easier scene questions, but their performance rapidly drops on the harder reasoning questions. On average, consistency falls from $83.3\%$ or $60.6\%$ between the tasks. For some models, a well-above-chance consistency drops a near-random baseline for

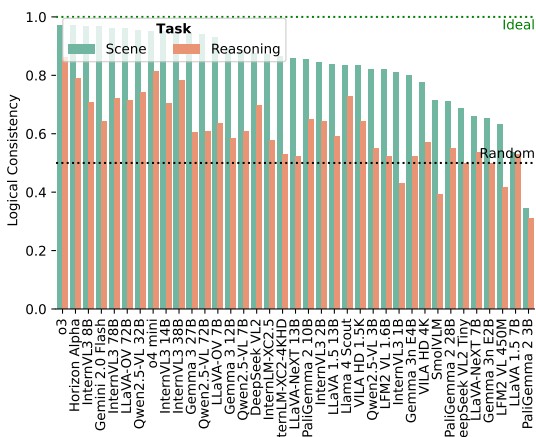

Figure 4: **Logical consistency.**

reasoning, suggesting that models are now guessing independently of the context, while providing well-grounded answers on the original task. In some cases, we also find a well-below random chance consistency, suggesting that the model is relying on shortcuts for shortcuts rather than the visual inputs. Alarmingly, we find PaliGemma2 3B to be susceptible to these for both tasks.

## 5 RELATED WORK

**Large Multi-Modal Models.** Recent progress in VLMs has significantly advanced the integration of visual and linguistic modalities, enabling more sophisticated multi-modal understanding and generation. Early approaches connect pretrained vision encoders with large language models via lightweight modules, achieving competitive performance with relatively few trainable parameters (Li et al., 2023c; Zhang et al., 2023; Zhu et al., 2023). The LLaVA series (Liu et al., 2023b;a; 2024a; Li et al., 2024a) improves visual instruction tuning, demonstrating stronger performance on fine-grained visual tasks. More recent models extend these capabilities to multi-image contexts, enabling richer scene understanding and more coherent textual reasoning (Li et al., 2024a; Steiner et al., 2024; Zhu et al., 2025; Bai et al., 2025; Gemma Team, 2025). Proprietary VLMs, including GPT and the o-series (OpenAI, 2024; 2025), and Gemini (Gemini Team, 2024; 2025), further highlight progress in versatile, context-aware multimodal learning frameworks, sometimes even including multi-modal reasoning (OpenAI, 2025).

Despite these advancements, many VLMs still exhibit notable weaknesses in visual understanding. Prior work has shown that they struggle with counting (Paiss et al., 2023), spatial reasoning, concept binding, and dense scene understanding (Doveh et al., 2023a;b; Huang et al., 2024; Campbell et al., 2024), as well as detailed image classification tasks (Mirza et al., 2025; 2023; Zhang et al., 2024b). In our work, we build on these findings by introducing a benchmark of densely populated public-domain paintings, designed to probe such vulnerabilities and evaluate the capacity of VLMs to perform basic visual tasks in challenging, visually overloaded scenes.

**Multi-Modal Vision Benchmarks.** The rapid progress of VLMs has spurred a surge of benchmarks evaluating their ability to integrate vision and language across tasks such as VQA, captioning, reasoning, and instruction following. Extending classic VQA datasets (Antol et al., 2015; Goyal et al., 2017), modern benchmarks vary in scope, from real-world instruction following in VisitBench (Bitton et al., 2023) to conversational reasoning in LLaVA-Bench (Bordes et al., 2024), zero-shot capability assessment across 16 capabilities, including OCR and spatial reasoning in MMVet (Yu et al., 2024), and multiple-choice probing in 12 dimensions in SeedBench (Li et al., 2023b). Broader frameworks such as MM-Bench (Liu et al., 2024b), TouchStone (Bai et al., 2023), OmniBench (Li et al., 2024b), and MMStar (Chen et al., 2024) aim for holistic multimodal evaluation by covering a wide array of tasks and domain-specific knowledge. MMMU (Yue et al., 2024) pushes toward expert-level multimodal reasoning. As performance on most of these benchmarks seems to saturate, more carefully designed benchmarks (Wu et al., 2023; Huang et al., 2024; Thrush et al., 2022; Hsieh et al., 2023) reveal persistent weaknesses in multiple dimensions, highlighting a discrepancy between many seemingly positive benchmark results and actual visual capabilities.

While these efforts nonetheless provide valuable insights, most emphasize global understanding, a very broad task coverage, or require domain-specific expertise, while often overlooking basic perception in more challenging settings, such as visually overloaded scenes. Recently, multiple benchmarks started the exploration of small details in high-resolution scenes (Wu & Xie, 2024; Li et al., 2023a; Shi et al., 2025), showing another hurdle in the development of vision models. Our work complements these benchmarks with VisualOverload, a human-annotated dataset of VQA pairs grounded in high-resolution, densely populated artworks. A key differentiator of high-resolution benchmarks is that VisualOverload aims at exploiting the full complexity of the scene, while previous works mostly model needle-in-the-haystack-style retrieval of small details. By focusing on six basic tasks in overloaded scenes, VisualOverload reveals systematic error modes in state-of-the-art open and proprietary VLMs, highlighting critical gaps in knowledge-free visual understanding.

## 6 CONCLUSION

In this work, we introduced VisualOverload, a novel VQA benchmark designed to expose the limitations of state-of-the-art VLMs in complex, detail-rich scenes. Our findings demonstrate that while these models perform well on global tasks, they consistently struggle with simple, fine-grained questions within visually "overloaded" environments. This performance gap highlights a critical area for future research, suggesting that the problem of fundamental visual understanding is far from solved. Ultimately, our dataset offers a crucial resource for the community to develop more robust and perceptive VLMs.

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

# A APPENDIX

## A.1 BENCHMARK PROMPTS

We used the following prompts in our main evaluation, depending on the question type (multiple-choice, counting, or OCR):

---

**Default Prompt for Multiple-Choice Questions**

`{Question}` Options:
A. `{Option A}`
B. `{Option B}`
...
Answer with the option's letter from the given choices directly.

---

**Default Prompt for OCR Questions**

`{Question}` Answer directly.

---

**Default Prompt for Counting Questions**

`{Question}` Answer with a number directly.

---

## A.2 ABLATION OF RESOLUTION

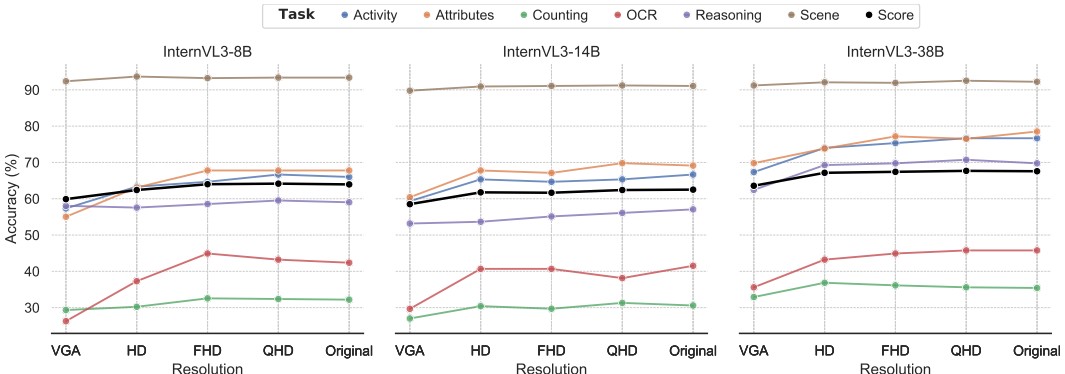

Figure 5: **Resolution ablation.**

We distribute VisualOverload at a resolution that matches the pixels of 4K (with a few outliers). Additionally, we downsampled images to match the number of pixels of VGA ($640 \times 480$ pixels), HD ($1280 \times 720$ pixels), FHD ($1920 \times 1080$ pixels), QHD ($2560 \times 1440$ pixels), and measured task-level performances on various instances of InternVL3 models in comparison to our original resolution. The results are shown in Fig. 5.

Generally, performance improves with resolution, but at a minor rate. However, it is visible that improvements are differently correlated with tasks. Text (especially small one) is poorly compressible, and it is, thus, unsurprising to see a strong correlation between resolution and OCR performance. The opposite is modeled by scene recognition, which, for the most part, is solvable by global features that should be detectable even at extreme compression. This is backed by the lack of significant performance deviation throughout our tested resolutions. For the other tasks, we typically see an increase in performance with resolution, which seems to plateau after Full HD resolution.

This is likely not a shortcoming of our benchmark, but rather attributed to the model's architecture. By default, InternVL3 splits the input image into at most 12 patches (each $448 \times 448$ pixels) plus a thumbnail (Zhu et al., 2025). Thus, the model only supports a resolution slightly above FHD without downsampling. While it is possible to increase the number of patches, this significantly increases the inference time and memory. For instance, even for InternVL3-8B, increasing the number of

patches from 12 to 40, which should be sufficient to process VisualOverload without downsampling, requires $8 \times 40$ GB GPUs, instead of just one, making such an experiment impossible for us. In theory, we, however, expect model performance to scale with resolution, assuming no downsampling. Consequently, we also expect higher performance using more patches (assuming a sufficient context window and proper training).

## A.3 LANGUAGE BIAS DETECTION

We use Gemini 2.5 Pro with the following prompt to detect language bias:

---
**Prompt for Language Bias Detection (Gemini 2.5 Pro)**

```
Below you will find a CSV with an excerpt of questions from a visual
question answering benchmark.  The benchmark is supposed to be only
solvable by looking at the image, however for the questions below,
most models are able to guess the correct option (ground_truth).  Your
task is to look at each questions, the options, and ground_truth and
to determine if the models were just lucky or there is some kind of
shortcut or language bias.  Provide an answer and rationale for each
question_id.

question_id, question, options, ground_truth
{CSV}
```
---

## A.4 PERFORMANCE WITH ADVANCED PROMPTING

Our evaluation in Sec. 3 utilizes simple prompts. In this section, we additionally ablate zero-shot chain-of-thought (CoT) (Wei et al., 2022; Kojima et al., 2022) on InternVL3 8B, the strongest 8B model on our benchmark, and an overall strong model. To this end, we modified the prompts as follows:

---
**CoT Prompt for Multiple-Choice Questions**

{Question} Options:
A. {Option A}
B. {Option B}
...
Think step by step. Answer with the option's letter from the given choices wrapped in <answer></answer>.

---
**CoT Prompt for OCR Questions**

{Question} Think step by step. Answer with the extracted text wrapped in <answer></answer>

---
**CoT Prompt for Counting Questions**

{Question} Think step by step. Answer with a number wrapped in <answer></answer>
---

The results in Tab. 3 show that at least for this model, CoT decreased performance on average. However, it significantly improved performance on the hardest split and for OCR. Since CoT prompting is primarily effective in large-scale LLMs (Wei et al., 2022), we hypothesize that the tested LLM may have been too small to benefit from CoT.

Table 3: **Comparison with CoT prompting.**

| Model | Params [B] | Activity (150) | Attributes (149) | Counting (559) | OCR (118) | Reasoning (356) | Scene (1388) | Easy (986) | Medium (1304) | Hard (430) | Total (2720) |
|---|---|---|---|---|---|---|---|---|---|---|---|
| InternVL3 38B (Zhu et al., 2025) | 38 | 76.7 | 78.5 | 35.4 | 45.8 | 69.8 | 92.2 | 99.7 | 81.8 | 7.2 | 67.6 |
| + CoT | 38 | 74.0 | 69.8 | 34.5 | 50.0 | 62.4 | 91.4 | 98.9 | 77.1 | 14.4 | 65.5 |

## A.5 EMBEDDING SPACE OF BENCHMARK QUESTIONS

We show a UMAP (McInnes et al., 2018) reduced embedding generated by Qwen3-embedding-4B (Zhang et al., 2025) of all questions (without answers) colored by task in Fig. 6. A clear separation of tasks is visible, except for the reasoning task, which overlaps with multiple other tasks as intended. The OCR questions form the most disconnected cluster.

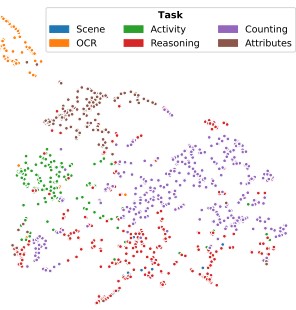

Figure 6: **Question Embeddings.**

## A.6 DATASHEET

In the following, we provide a datasheet (Gebru et al., 2021). We have anonymized some entries for the review process and will update these upon release.

| **Motivation** |
|:---:|

**For what purpose was the dataset created?** Was there a specific task in mind? Was there a specific gap that needed to be filled? Please provide a description.

 VisualOverload was created to test basic visual recognition skills of VLMs in densely populated scenes, as most prior VQA datasets often probe skills of superficial features.

**Who created this dataset (e.g., which team, research group) and on behalf of which entity (e.g., company, institution, organization)?**

 *(hidden during the review)*

**Who funded the creation of the dataset?** If there is an associated grant, please provide the name of the grantor and the grant name and number.

 *(hidden during the review)*

| **Composition** |
|:---:|

**What do the instances that comprise the dataset represent (e.g., documents, photos, people, countries)?** Are there multiple types of instances (e.g., movies, users, and ratings; people and interactions between them; nodes and edges)? Please provide a description.

 The dataset consists of images associated with multiple questions.

**How many instances are there in total (of each type, if appropriate)?**

 The dataset consists of 150 images and a total of 2720 questions.

**Does the dataset contain all possible instances or is it a sample (not necessarily random) of instances from a larger set?** If the dataset is a sample, then what is the larger set? Is the

sample representative of the larger set (e.g., geographic coverage)? If so, please describe how this representativeness was validated/verified. If it is not representative of the larger set, please describe why not (e.g., to cover a more diverse range of instances, because instances were withheld or unavailable).

The images are a subset of public domain artworks hosted on https://artsandculture.google.com filtered to display visually complex and dense scenes.

**What data does each instance consist of? "Raw" data (e.g., unprocessed text or images) or features?** In either case, please provide a description.

Each sample is a collection of the following items:

- `question_id`: Unique identifier of each question.
- `image`: A PIL JPEG image. Most of our images were resized to match the total pixel count of 4k (3840x2160 px) in different aspect ratios.
- `question`: A question about the image.
- `question_type`: Type of question. Will be one of choice (response expected to be "A", "B", "C", or "D"), counting (freeform), or ocr (freeform). You can use this information to request a suitable output format.
- `options`: This is the list of options for question_type=choice and empty otherwise. Please treat the options as answer options A, B, C, D (4 options) or A, B (2 options).
- `difficulty`: Meta-data about the difficulty of the question. One of easy, medium, or hard.
- `category`: Meta-data about the question task. One of activity, attributes, counting, ocr, reasoning, or scene.
- `default_prompt`: You can use this prompt to stay compliant with our results. It is a simple combination of the question and answers, with some additional output format constraints. This should work well for most models.

**Is there a label or target associated with each instance?** If so, please provide a description.

Each question is associated with a ground-truth. This ground-truth is hidden from the public to avoid test leakage.

**Is any information missing from individual instances?** If so, please provide a description, explaining why this information is missing (e.g., because it was unavailable). This does not include intentionally removed information, but might include, e.g., redacted text.

We obfuscate image file names and question IDs to reduce knowledge priors.

**Are relationships between individual instances made explicit (e.g., users' movie ratings, social network links)?** If so, please describe how these relationships are made explicit.

The samples in the dataset shall be treated independently.

**Are there recommended data splits (e.g., training, development/validation, testing)?** If so, please provide a description of these splits, explaining the rationale behind them.

All the samples in our dataset shall be exclusively treated as a test set. We do not provide development sets, as we consider all questions to be solvable with a basic set of skills that should be present in frontier VLMs.

**Are there any errors, sources of noise, or redundancies in the dataset?** If so, please provide a description.

All questions and ground truths are manually annotated and, thus, may contain errors. To reduce the error rate, we double-checked all questions where multiple models provided wrong answers.

**Is the dataset self-contained, or does it link to or otherwise rely on external resources (e.g., websites, tweets, other datasets)?** If it links to or relies on external resources, a) are there guarantees that they will exist, and remain constant, over time; b) are there official archival versions of the complete dataset (i.e., including the external resources as they existed at the time the dataset was created); c) are there any restrictions (e.g., licenses, fees) associated with any of the external resources that might apply to a future user? Please provide descriptions of all external resources and any restrictions associated with them, as well as links or other access points, as appropriate.

The dataset is self-contained.

**Does the dataset contain data that might be considered confidential (e.g., data that is protected by legal privilege or by doctor-patient confidentiality, data that includes the content of individuals non-public communications)?** If so, please provide a description.

No.

**Does the dataset contain data that, if viewed directly, might be offensive, insulting, threatening, or might otherwise cause anxiety?** If so, please describe why.

The dataset contains samples that show religious beliefs, (partial) nudity, and/or injury and death.

**Does the dataset relate to people?** If not, you may skip the remaining questions in this section.

The dataset contains artworks that may depict people.

**Does the dataset identify any subpopulations (e.g., by age, gender)?** If so, please describe how these subpopulations are identified and provide a description of their respective distributions within the dataset.

The dataset does not identify any subpopulations.

**Is it possible to identify individuals (i.e., one or more natural persons), either directly or indirectly (i.e., in combination with other data) from the dataset?** If so, please describe how.

Some of the individuals are of historical, biblical, or mythical origin and may be identified. No living individuals can be identified from the dataset.

**Does the dataset contain data that might be considered sensitive in any way (e.g., data that reveals racial or ethnic origins, sexual orientations, religious beliefs, political opinions or union memberships, or locations; financial or health data; biometric or genetic data; forms of government identification, such as social security numbers; criminal history)?** If so, please provide a description.

No.

| Collection Process |
| --- |

**How was the data associated with each instance acquired?** Was the data directly observable (e.g., raw text, movie ratings), reported by subjects (e.g., survey responses), or indirectly inferred/derived from other data (e.g., part-of-speech tags, model-based guesses for age or language)? If data was reported by subjects or indirectly inferred/derived from other data, was the data validated/verified? If so, please describe how.

Please see Sec. 2.

**What mechanisms or procedures were used to collect the data (e.g., hardware apparatus or sensor, manual human curation, software program, software API)?** How were these mechanisms or procedures validated?

Please see Sec. 2.

**If the dataset is a sample from a larger set, what was the sampling strategy (e.g., deterministic, probabilistic with specific sampling probabilities)?**

n/a.

**Who was involved in the data collection process (e.g., students, crowdworkers, contractors) and how were they compensated (e.g., how much were crowdworkers paid)?**

The dataset was collected and annotated by the authors of this paper. No crowdworkers, students, or contractors, *etc.*, were involved.

**Over what timeframe was the data collected? Does this timeframe match the creation timeframe of the data associated with the instances (e.g., recent crawl of old news articles)?** If not, please describe the timeframe in which the data associated with the instances was created.

The images were collected between April and May 2025, and annotated and cleaned between May and August 2025.

**Were any ethical review processes conducted (e.g., by an institutional review board)?** If so, please provide a description of these review processes, including the outcomes, as well as a link or other access point to any supporting documentation.

No.

**Does the dataset relate to people?** If not, you may skip the remaining questions in this section.

The dataset contains artworks that may depict people.

**Did you collect the data from the individuals in question directly, or obtain it via third parties or other sources (e.g., websites)?**

n/a.

**Were the individuals in question notified about the data collection?** If so, please describe (or show with screenshots or other information) how notice was provided, and provide a link or other access point to, or otherwise reproduce, the exact language of the notification itself.

All depicted individuals are no longer alive.

**Did the individuals in question consent to the collection and use of their data?** If so, please describe (or show with screenshots or other information) how consent was requested and provided, and provide a link or other access point to, or otherwise reproduce, the exact language to which the individuals consented.

n/a.

**If consent was obtained, were the consenting individuals provided with a mechanism to revoke their consent in the future or for certain uses?** If so, please provide a description, as well as a link or other access point to the mechanism (if appropriate).

n/a.

**Has an analysis of the potential impact of the dataset and its use on data subjects (e.g., a data protection impact analysis) been conducted?** If so, please provide a description of this analysis, including the outcomes, as well as a link or other access point to any supporting documentation.

n/a.

---

### Preprocessing/cleaning/labeling

**Was any preprocessing/cleaning/labeling of the data done (e.g., discretization or bucketing, tokenization, part-of-speech tagging, SIFT feature extraction, removal of instances, processing**

**of missing values)?** If so, please provide a description. If not, you may skip the remainder of the questions in this section.

Yes, see Sec. 2.

**Was the "raw" data saved in addition to the preprocessed/cleaned/labeled data (e.g., to support unanticipated future uses)?** If so, please provide a link or other access point to the "raw" data.

The raw data can be requested from the authors.

**Is the software used to preprocess/clean/label the instances available?** If so, please provide a link or other access point.

The images were obtained using `https://github.com/lovasoa/dezoomify-rs`. All further processing scripts were developed by the authors and are not available publicly.

---

| Uses |
|:---:|

**Has the dataset been used for any tasks already?** If so, please provide a description.

The dataset has been used to evaluate basic visual skills of frontier VLMs in Sec. 3.

**Is there a repository that links to any or all papers or systems that use the dataset?** If so, please provide a link or other access point.

*(hidden during the review)*

**What (other) tasks could the dataset be used for?**

The dataset is primarily designed for visual question answering (VQA), but we encourage users to apply it to other tasks as desired.

**Is there anything about the composition of the dataset or the way it was collected and preprocessed/cleaned/labeled that might impact future uses?** For example, is there anything that a future user might need to know to avoid uses that could result in unfair treatment of individuals or groups (e.g., stereotyping, quality of service issues) or other undesirable harms (e.g., financial harms, legal risks) If so, please provide a description. Is there anything a future user could do to mitigate these undesirable harms?

No.

**Are there tasks for which the dataset should not be used?** If so, please provide a description.

This dataset is released exclusively for academic research and educational use. It must not be applied to purposes that could lead to harm, including surveillance, discrimination, exploitation, harassment, or the generation of misleading or offensive content. Users are expected to uphold the highest standards of research integrity and ethics, and to ensure that their work with this dataset aligns with responsible AI principles.

---

| Distribution |
|:---:|

**Will the dataset be distributed to third parties outside of the entity (e.g., company, institution, organization) on behalf of which the dataset was created?** If so, please provide a description.

**How will the dataset will be distributed (e.g., tarball on website, API, GitHub)** Does the dataset have a digital object identifier (DOI)?

**When will the dataset be distributed?**

The dataset is primarily distributed through: *(hidden during the review)*.

**Will the dataset be distributed under a copyright or other intellectual property (IP) license, and/or under applicable terms of use (ToU)?** If so, please describe this license and/or ToU, and provide a link or other access point to, or otherwise reproduce, any relevant licensing terms or ToU, as well as any fees associated with these restrictions.

The dataset is distributed under the Creative Commons Attribution-ShareAlike 4.0 International license without any further terms of use.

**Have any third parties imposed IP-based or other restrictions on the data associated with the instances?** If so, please describe these restrictions, and provide a link or other access point to, or otherwise reproduce, any relevant licensing terms, as well as any fees associated with these restrictions.

No.

**Do any export controls or other regulatory restrictions apply to the dataset or to individual instances?** If so, please describe these restrictions, and provide a link or other access point to, or otherwise reproduce, any supporting documentation.

No.

---

| Maintenance |
|:-----------:|

**Who will be supporting/hosting/maintaining the dataset?**

The authors will be supporting/hosting/maintaining the dataset.

**How can the owner/curator/manager of the dataset be contacted (e.g., email address)?**

The authors can be contacted via GitHub issue at: *(hidden during the review)*.

**Is there an erratum?** If so, please provide a link or other access point.

No.

**Will the dataset be updated (e.g., to correct labeling errors, add new instances, delete instances)?** If so, please describe how often, by whom, and how updates will be communicated to users (e.g., mailing list, GitHub)?

The dataset will not be modified to ensure comparability of results. Corrected or derived datasets will be released independently.

**If the dataset relates to people, are there applicable limits on the retention of the data associated with the instances (e.g., were individuals in question told that their data would be retained for a fixed period of time and then deleted)?** If so, please describe these limits and explain how they will be enforced.

n/a.

**Will older versions of the dataset continue to be supported/hosted/maintained?** If so, please describe how. If not, please describe how its obsolescence will be communicated to users.

The dataset will remain available as long as it continues to be hosted by the third-party platforms on which it is stored.

**If others want to extend/augment/build on/contribute to the dataset, is there a mechanism for them to do so?** If so, please provide a description. Will these contributions be validated/verified? If

so, please describe how. If not, why not? Is there a process for communicating/distributing these contributions to other users? If so, please provide a description.

Users can extend/augment/build upon the dataset, but must publish their new work as a standalone derivative. We kindly request that users communicate any releases to the authors.

