# OpenReview forum: "VisualOverload: Probing Visual Understanding of VLMs in Really Dense Scenes"
_ICLR.cc/2026/Conference — ICLR 2026 Conference Withdrawn Submission_

### Official Review · Reviewer_Hz6e · 2025-10-24

**Soundness:** 2
**Presentation:** 3
**Contribution:** 2
**Rating:** 2
**Confidence:** 4

**Summary:**

This paper presents a new benchmark for evaluating VLMs focusing on activity, reasoning, OCR, scene, attribute, and counting tasks for dense images/paintings (i.e. complex images containing many objects). These are basic perception tasks that VLLMs should be well-performing.

**Strengths:**

- the idea of finding failure models of VLMs is interesting and of value to the community

**Weaknesses:**

- It seems to me that the paper results contradict the paper claims. From Table 2 we can see pretty decent performance of VLLMs for easy and medium tasks, also scaling with model size. There's a hard set where most of the VLLMs don't do well but i think even in well established benchmarks there are test samples that the models don't work.
- Some issues identified in the paper (e.g. counting, attributes, OCR) are already known  to be existing issues with current VLLMs.
- The contribution of such a small size benchmark is not enough for ICLR

**Questions:**

N/A

---

### Official Review · Reviewer_48xJ · 2025-10-29

**Soundness:** 3
**Presentation:** 2
**Contribution:** 2
**Rating:** 4
**Confidence:** 3

**Summary:**

This paper introduces VisualOverload, a Visual Question Answering (VQA) benchmark designed to test state-of-the-art Vision-Language Models (VLMs) in dense, high-resolution scenes—addressing a gap in existing benchmarks that overestimate VLMs’ visual understanding by focusing on global or simple tasks.  The benchmark comprises 150 public-domain high-resolution paintings (mostly ~4K resolution) with dense, detail-rich scenes, paired with 2,720 manually annotated question-answer pairs across 6 tasks: activity recognition, attribute recognition, counting, OCR, reasoning, and scene classification. Binary questions are paired with logical opposites to measure consistency, and questions are split into easy/medium/hard based on model performance; ground truths are private to avoid leakage.  Evaluations of 37 VLMs (open-weight: 0.4B–109B params; proprietary: e.g., o3, Gemini 2.0 Flash) show even the top model (o3) achieves only 19.6% accuracy on the hard split and 69.5% overall. VLMs struggle most with counting (top accuracy: 41.7%) and OCR (top accuracy: 62.7%), and exhibit logical inconsistencies in reasoning tasks.

**Strengths:**

1. The paper introduces the VisualOverload benchmark, which specifically focuses on testing VLMs’ fine-grained visual understanding in dense, high-resolution scenes—an area where existing VQA benchmarks (which prioritize global scene understanding or simple retrieval tasks) fail. By using 150 public-domain high-resolution paintings (mostly ~4K) with manually annotated questions across 6 core tasks (e.g., counting, OCR, reasoning) and pairing binary questions with logical opposites, the benchmark effectively exposes VLMs’ limitations in handling real-world complex visual scenarios, filling a key gap in VLM evaluation
2. The study evaluates 37 diverse VLMs (covering open-weight models of varying sizes and proprietary frontier models) and uses a well-calibrated difficulty split (easy/medium/hard based on average model performance) to ensure meaningful performance differentiation. Additionally, its detailed error analysis uncovers systematic VLM failures—such as poor counting accuracy for large numbers, severe OCR errors (high normalized Levenshtein edit distances), and logical inconsistencies in paired questions—providing actionable insights for improving VLM design, rather than just reporting performance metrics

**Weaknesses:**

1. Insufficient Validation of Benchmark Generalization and Fairness: The benchmark’s image corpus is limited to 150 public-domain paintings, which are culturally specific (curated from museum collections) and may not represent the diversity of real-world dense scenes (e.g., urban crowds, medical imaging, industrial layouts). The paper does not evaluate whether VisualOverload’s findings generalize to non-artwork dense scenarios, raising concerns about its external validity. Additionally, while it mentions removing language biases (Section 1-51), it does not assess potential cultural biases in annotations (e.g., questions tailored to Western art contexts) or how these might impact model performance across diverse VLMs trained on different data.
2. The paper’s discussion of related work (Section 5) remains relatively superficial, failing to deeply contextualize VisualOverload against recent high-resolution or dense-scene VQA benchmarks (e.g., Wu & Xie, 2024; Shi et al., 2025). It only briefly mentions that prior high-resolution benchmarks focus on "needle-in-the-haystack" retrieval rather than full-scene complexity but lacks a rigorous, side-by-side comparison of benchmark design (e.g., image source, task scope, annotation methodology) or quantitative performance contrasts with these existing datasets.

**Questions:**

The benchmark’s image corpus is limited to 150 public-domain paintings, which are mostly from museum collections and may have cultural specificity. Could you elaborate on whether there are plans to expand the image sources to include more diverse real-world dense scenes (such as urban crowds, medical images, or industrial layouts) in the future? And how do you intend to verify whether the limitations of VLMs revealed by the current painting-based benchmark can be generalized to these non-artwork dense scenarios?

---

### Official Review · Reviewer_1ZXV · 2025-10-30

**Soundness:** 3
**Presentation:** 3
**Contribution:** 3
**Rating:** 4
**Confidence:** 4

**Summary:**

This paper introduces a VQA benchmark in dense, high-resolution (visually overloaded) scenes with 2,720 manually curated question–answer pairs over activity recognition, attribute recognition, counting, OCR, visual reasoning, and global scene classification tasks. The evaluation results show that the sate-of-the-art models struggle significantly in fine-grained understanding in dense settings, particularly for counting and OCR. The analysis of this paper finds that systematic inconsistencies and shortcut biases hinder robust performance in visually overloaded settings.

**Strengths:**

1. The images are new from existing dataset with high resolution and fine-grained details, and annotations are labeled manually.
2. The quality of the data is well controlled, while 37 VLMs are evaluated on proposed dataset and manually verified the correctness of ground truths.

**Weaknesses:**

1. It is a consensus within the community that current VLMs have shortcomings in OCR and counting tasks, while this paper lacks discussions comparing related benchmarks focusing on these tasks to demonstrate the novelty to propose VisualOverload.
2. Some results of state-of-the-art models are missing, such as GPT-4o and Gemini-2.5 Pro.
3. The image are collected from Google Arts & Culture, the evaluation field of images may be limited.

**Questions:**

1. Does the conclusion in Section 4 hold consistent across the evaluation results of all the state-of-the-art models, or does it only apply to some of the current models? Are there some differences in the performance over these tasks of different models within the community?
2. For easy questions, as shown in Table 2, nearly all models achieve above 98 points. Is this easy part of the dataset necessary?

---

### Official Review · Reviewer_Mb4w · 2025-11-01

**Soundness:** 3
**Presentation:** 3
**Contribution:** 3
**Rating:** 4
**Confidence:** 4

**Summary:**

This paper introduces VisualOverload, a new benchmark designed to evaluate visual question answering (VQA) models, specifically targeting their performance in densely populated, high-resolution scenes. The aim is to probe basic visual understanding in models, particularly their ability to handle fine-grained recognition in complex scenes. The authors identify that while state-of-the-art vision-language models (VLMs) perform well on simpler VQA tasks, they struggle with these more nuanced, visually overloaded contexts, revealing significant weaknesses.

**Strengths:**

- The proposed dataset and the specific focus on dense, high-resolution images are innovative. The focus on challenging tasks like counting and OCR in complex settings provides important insights into the limitations of current VLMs.
- The evaluation of 37 models is thorough.

**Weaknesses:**

- Although the paper discusses the impact of resolution on model performance, the analysis is relatively simple. Different models may have varying processing capabilities to respond differently to high-resolution images, possibly due to differences in the amount of pre-trained data; the paper does not delve into these architectural differences.
- The error analysis section reveals the model's failure modes in tasks such as counting, OCR, and logical consistency. However, this analysis mainly stays at the task level and lacks in-depth analysis of the root causes of model failures.
- While the paper reveals the limitations of current VLMs in complex scenarios, it lacks specific suggestions on how to improve these models. For example, what problems were discovered that could provide insights for the future development of these models?
- While Figure 1 shows an example problem and corresponding image for one task, there aren't enough images to demonstrate the diversity of the dataset. Several example images for different task categories could be added, showing how these images are designed and how different tasks combine these images to evaluate model performance.
- The difficulty level of the dataset (easy, medium, hard) is an important point of analysis in the paper, but there is currently no clear visualization in the paper to show how these difficulty levels affect the model's performance.

**Questions:**

Please see the weaknesses.

---

### Official Review · Reviewer_sMgK · 2025-11-02

**Soundness:** 2
**Presentation:** 2
**Contribution:** 2
**Rating:** 4
**Confidence:** 3

**Summary:**

The authors develop VisualOverload, a visual question answering (VQA) benchmark visual language models (VLMs) on a focus of visually dense images. The paper motivates this by proposing a possible bottleneck in modern VLMs, namely the visual encoder, which may limit the amount of information that can be fit in a fixed set of tokens, and argue VisualOverload helps test this. The authors collect 150 photoscans of real artworks from Google Arts & Culture and develop 2k+ questions, including counting, global scene understanding, and OCR -type tasks. The authors find that the best performing model, o3, was able to only achieve 69.5% overall.

**Strengths:**

1. The research question is well-motivated. Many benchmarks really do tend to focus on global scene understanding. While visually dense images do occur in many benchmarks, to the best of my knowledge it is true that this has never been a central focus of any other benchmark to this date.
2. The research hypothesis that the vision encoder is a bottleneck is really interesting. It's definitely a point of concern, and a study on that hypothesis would be very useful.
3. The authors evaluate on a comprehensive set of vision language models.
4. The authors clearly put significant effort in the curation of the images and high-quality questions.

**Weaknesses:**

1. In the introduction, the authors claim a motivation to test visually dense scenes is a hypothesis that the vision encoder is a bottleneck for VLMs. However, their experimental setup does not do a good job of directly testing this:
	* While encoders might compress images to a fixed number of tokens, not all *VLMs* encode into a fixed number of tokens.
		* Authors use o3 as an example of a frontier model that does poorly - o3 has an option to use either low fidelity or [high fidelity](https://platform.openai.com/docs/guides/images-vision?api-mode=responses#gpt-4o-gpt-4-1-gpt-4o-mini-cua-and-o-series-except-o4-mini)
		* InternVL3 also uses [dynamic resolution](https://internvl.github.io/blog/2025-04-11-InternVL-3.0/)
		* This would have been an interesting experimental setup to compare fixed resolution performance vs dynamic resolution
	* Error analysis and conclusion do not study whether or not visual cluttering was the cause of errors.
2. The paper lacks a formal definition of image complexity as a metric for selection. While the authors state that it is hard to formally define it, an alternative that might be useful would be some statistics about the number of subjects or objects in each photo, and compare it against other VQA benchmarks in a more quantitative matter.
3. Some more clarification on how the easy/medium/hard test splits were created would be helpful.
   "We divide our questions into three difficulty levels—easy, medium, and hard—based on model performance in Sec. 3. The thresholds are defined by the percentage of correct responses: \[0, 20\] for hard, (20, 90) for medium, and \[90, 100\] for easy."
	* Does this mean:
		* a) if average model performance on a question is lower than 20%, then the question is in the hard split, or
		* b) hard test split is designed so that all models perform <= 20%.
	* In either case, this makes the claim that "even the best model (o3) … only achieves 19.8% accuracy on our hardest test split" somewhat circular.
4. Lack of image diversity:
	* Despite a focus on visual clutterness, the paper also happens to only include historical artworks, which may result in an important confounding variable during evaluation.
	* Most paintings also have humans as the main subjects. While not as large of a concern, some statistics about the dataset and its subjects would also have been useful.
5. The claims of comparisons against existing benchmarks are not substantiated:
   "Existing VQA benchmarks underestimate the true difficulty of visual reasoning. They rely on low-resolution images, recycled content, and automatically generated questions that encourage shallow pattern matching rather than genuine scene understanding"
	* Do you mean VQA benchmarks as in the VQA v1/VQA v2 *datasets/benchmarks*, and not VQA as a task? This is more than a bit unfair to others' works on VQA *tasks*. For example, the below are more recent VQA benchmarks that use high resolution photos and is not "recycling" COCO data. Just a few examples
		* [HRVQA](https://arxiv.org/abs/2301.09460): high-resolution images (1024x1024).
		* [DocVQA](https://openaccess.thecvf.com/content/WACV2021/html/Mathew_DocVQA_A_Dataset_for_VQA_on_Document_Images_WACV_2021_paper.html): most photos have more than [1.68k px](https://huggingface.co/datasets/lmms-lab/DocVQA) in width.
		* [SlideVQA](https://huggingface.co/datasets/NTT-hil-insight/SlideVQA): larger than 1k px in width.
		* [Blink](https://huggingface.co/datasets/BLINK-Benchmark/BLINK): very large images, high quality questions, not recycled content.
	* This should be completely reworded and have a fair comparison.

**Questions:**

1. Could you clarify how the easy/medium/hard test splits were created?

---

### Note · Authors · 2025-11-13

I have read and agree with the venue's withdrawal policy on behalf of myself and my co-authors.